# Bisphenols A and S Alter the Bioenergetics and Behaviours of Normal Urothelial and Bladder Cancer Cells

**DOI:** 10.3390/cancers14164011

**Published:** 2022-08-19

**Authors:** Ève Pellerin, Félix-Antoine Pellerin, Stéphane Chabaud, Frédéric Pouliot, Stéphane Bolduc, Martin Pelletier

**Affiliations:** 1Centre de Recherche en Organogénèse Expérimentale/LOEX, Regenerative Medicine Division, CHU de Québec-Université Laval Research Center, Quebec, QC G1J 1Z4, Canada; 2Oncology Division, CHU de Québec-Université Laval Research Center, Quebec, QC G1R 2J6, Canada; 3Department of Surgery, Faculty of Medicine, Laval University, Quebec, QC G1V 0A6, Canada; 4Infectious and Immune Disease Division, CHU de Québec-Université Laval Research Center, Quebec, QC G1V 4G2, Canada; 5Department of Microbiology-Infectious Diseases and Immunology, Faculty of Medicine, Laval University, Quebec, QC G1V 0A6, Canada

**Keywords:** bladder cancer, bisphenol A, bisphenol S, energy metabolism, migration, proliferation

## Abstract

**Simple Summary:**

This research brings new knowledge on the potential roles of bisphenol A and bisphenol S on bladder cancer progression. By assessing the impact of bisphenols A and S on normal urothelial cells and non-invasive and invasive bladder cancer cells, this study aimed to demonstrate that these endocrine-disrupting chemicals could promote bladder cancer progression through the alteration of the bioenergetics and behaviours of healthy and cancerous bladder cells. These results could provide a better understanding of the pathophysiology of bladder cancer and its hormone-sensitive characteristics. Furthermore, this study suggests that bisphenols A and S could affect bladder cancer recurrence, progression and patient prognosis.

**Abstract:**

Bisphenol A (BPA) and bisphenol S (BPS) are used in the production of plastics. These endocrine disruptors can be released into the environment and food, resulting in the continuous exposure of humans to bisphenols (BPs). The bladder urothelium is chronically exposed to BPA and BPS due to their presence in human urine samples. BPA and BPS exposure has been linked to cancer progression, especially for hormone-dependent cancers. However, the bladder is not recognized as a hormone-dependent tissue. Still, the presence of hormone receptors on the urothelium and their role in bladder cancer initiation and progression suggest that BPs could impact bladder cancer development. The effects of chronic exposure to BPA and BPS for 72 h on the bioenergetics (glycolysis and mitochondrial respiration), proliferation and migration of normal urothelial cells and non-invasive and invasive bladder cancer cells were evaluated. The results demonstrate that chronic exposure to BPs decreased urothelial cells’ energy metabolism and properties while increasing them for bladder cancer cells. These findings suggest that exposure to BPA and BPS could promote bladder cancer development with a potential clinical impact on bladder cancer progression. Further studies using 3D models would help to understand the clinical consequences of this exposure.

## 1. Introduction

In the last decades, hormone-dependent cancers, such as breast [1] and prostate cancers [2], have increased in industrialized countries. Among several causes, part of this rise could be associated with the growing number of endocrine disruptors found in the environment [3]. More than 350 synthetic molecules in the environment, including bisphenols (BPs), are considered endocrine disruptors because of their capacity to modulate the action or the metabolism of various hormones in the organism [3].

The bisphenol family comprises many molecules, such as bisphenol A (BPA) and bisphenol S (BPS). BPA is used to produce plastics, such as polycarbonate and epoxy resins [4]. It is found in various daily objects (e.g., water bottles and food containers) [5], making this compound ubiquitous in the environment [6]. Exposure to BPA has been associated with cancer development, especially in hormone-dependent cancers such as breast [7] and prostate cancers [8]. Recently, BPS has been used as a safer alternative to BPA in plastic production because of its excellent stability [9]. However, BPS has also been linked with cancer progression [7,10]. BPA is found at measurable concentrations in the urine of >90% of the population [10], whereas BPS is detectable in 78% of urine samples at comparable concentrations to BPA [11]. BPA and BPS are found at similar concentrations in urine, ranging from 1 to 100 nM [11]. The ubiquitous presence of BPA and BPS in the urine results in chronic exposure to the urinary tract, particularly the bladder, where urine can be stored for many hours [12].

The multiple binding capacities of BPA and BPS allow the compounds to alter different signalling pathways associated with cell migration, proliferation and invasion [10,13]. In addition, studies have shown that the bladder’s urothelial cells (UCs) express cell receptors targeted by BPA and BPS, such as estrogen receptors (ERs) α and β (ERα and ERβ), the androgen receptor (AR) and the G protein-coupled estrogen receptor (GPER) [14,15]. Although the bladder is not recognized as a hormone-sensitive tissue, studies have shown the role of these sex steroid receptors in bladder cancer initiation and progression. As such, the activation of ERα could promote the proliferation of bladder cancer cells [16], whereas ERβ and AR could promote bladder cancer growth and invasion via the alteration of tumour suppressor gene expression [17,18]. Therefore, the pro-tumorigenic tendencies of BPA and BPS, their binding capacities to cell receptors and the presence of these key receptors in the bladder urothelium could suggest a potential role for BPs in bladder cancer development [19].

It was previously demonstrated that the energy metabolism of healthy bladder fibroblasts decreased after chronic exposure to physiological concentrations of BPA. In contrast, BPA-exposed cancer-associated fibroblasts (CAFs) displayed an increased glycolytic metabolism, leading to extracellular acidification [20]. This enhanced acidification can lead to the inhibition of immune cells, such as monocytes, as well as reorganization of the extracellular matrix [21,22], thus potentially promoting bladder cancer progression [20]. The hypothesis of this study was that BPA and BPS would impact the energy metabolism and properties of normal urothelial cells and bladder cancer cells, i.e., migration, proliferation and expression of cell markers of invasive potential, which could promote bladder cancer initiation and progression.

## 2. Materials and Methods

### 2.1. Cell Lines

All procedures involving patients were conducted according to the Helsinki Declaration and were approved by the local Research Ethical Committee. Each specimen was obtained with the donor’s consent for tissue harvesting, and all experimental procedures were performed according to the CHU de Québec-Université Laval guidelines. Normal urothelial cells (UCs) were extracted from healthy human urological tissue biopsies and cultured as previously described [23,24]. The UCs were isolated from two healthy paediatric volunteers undergoing surgery for a benign condition (UC1 and UC2) and used as non-transformed primary cell lines.

UCs, RT4 non-invasive bladder cancer cells (ATCC HTB-2) and T24 invasive bladder cancer cells (ATCC HTB-4) were maintained in culture media composed of a 3:1 mix of Dulbecco–Vogt modification of Eagle’s medium (DMEM) (Invitrogen, Burlington, ON, Canada) and Ham F12 medium (Invitrogen) supplemented with 5% fetal bovine serum clone II (Hyclone, GE Healthcare Life Science, Wauwatosa, WI, USA), 24.3 µg/mL adenine (Corning, Tewksbury, MA, USA), 5 µg/mL insulin (Sigma-Aldrich, Oakville, ON, Canada), 0.212 µg/mL isoproterenol (Sandoz, Boucherville, QC, Canada), 0.4 mg/mL hydrocortisone (Calbiochem, San Diego, CA, USA), 10 ng/mL epidermal growth factor (Austral Biologicals, San Ramon, CA, USA), 100 U/mL penicillin (Sigma-Aldrich) and 25 mg/mL gentamicin (Schering-Plough Canada Inc./Merck, Scarborough, ON, Canada), and incubated at 37 °C with 8% CO_2_. Media were changed three times per week.

### 2.2. Seahorse Energy Metabolism Measurements

UCs, RT4 and T24 cells were plated in 96-well Seahorse XF cell culture plates (Agilent/Seahorse Bioscience, Chicopee, MA, USA) and exposed or not to 10^−8^ M BPA (Millipore Sigma, Oakville, ON, Canada) or 10^−8^ M BPS (Millipore Sigma) for 72 h before measurements. Media were changed every day. Seahorse XFe96 sensor cartridge plates (Agilent/Seahorse Bioscience) were hydrated the day before the analysis with the XF Calibrant (Agilent/Seahorse Bioscience) and incubated overnight at 37 °C without CO_2_. Before the bioenergetics measurements, cells were washed and incubated for one hour with Glyco Stress media or Mito Stress media. Glyco Stress media contained XF Base Medium (minimal DMEM) (Agilent/Seahorse Bioscience) supplemented with 2 mM L-glutamine (Wisent Bioproducts Inc., Saint-Jean-Baptiste, QC, Canada). Mito Stress media consisted of XF Base Medium supplemented with 2 mM L-glutamine, 1 mM sodium pyruvate (Wisent Bioproducts Inc.) and 10 mM D-(+)-glucose (Millipore Sigma). The extracellular acidification rate (ECAR), representative of glycolytic metabolism, and the oxygen consumption rate (OCR), representative of mitochondrial respiration, were determined using the XFe Extracellular Flux Analyzer (Agilent/Seahorse Bioscience) [20,25].

The glycolytic metabolism was established by the sequential injection of 10 mM D-(+)-glucose (Millipore Sigma), 1.5 µM of the ATP synthase inhibitor oligomycin (Cayman Chemical, Ann Arbor, MI, USA) to inhibit mitochondrial respiration and force the cells to maximize their glycolytic capacity, and 50 mM 2-deoxy-D-glucose (2-DG) (Alfa Aesar, Ward Hill, MA, USA), a competitive inhibitor of the first step of glycolysis.

The mitochondrial respiration was determined by the sequential injection of 1.5 µM of the ATP synthase inhibitor oligomycin (Cayman Chemical), 0.5 µM of the mitochondrial uncoupler trifluoromethoxy carbonylcyanide phenylhydrazone (FCCP) (Cayman Chemical) and a combination of 0.5 µM of the mitochondrial complex I inhibitor rotenone (MP Biomedicals, Santa Ana, CA, USA) and 0.5 µM of the mitochondrial complex III inhibitor antimycin A (Millipore Sigma). The concentrations indicated for each injection represent the final concentrations in the wells. At least three measurement cycles (3 min of mixing + 3 min of measuring) were completed before and after each injection.

The OCR and ECAR were calculated using Wave software v2.6 (Agilent/Seahorse Bioscience). Following the manufacturer’s instructions, energy metabolism was normalized according to the number of cells using a CyQuant Cell proliferation assay kit (Invitrogen). The fluorescence of each well was measured at 485 nm/535 nm for 0.1 s using the Victor2 1420 MultiLabel Counter plate reader (Perkin Elmer Life Sciences, Waltham, MA, USA) and Wallac 1420 software (Perkin Elmer). The normalization values were calculated from the fluorescence measurements with Microsoft Excel software (Microsoft, Redmond, WA, USA) and applied to the metabolic values. Metabolic values were presented as percentages with 100% established using the first three measurements. Therefore, the baseline was established before the injection of glucose or oligomycin (see Appendix A). Each experiment included at least three replicates per condition (n ≥ 3), and each experiment was repeated at least three times (N ≥ 3).

### 2.3. Proliferation

On day 0, UCs, RT4 and T24 cells were seeded in 24-well culture plates at 60,000 cells/well density. Cells were incubated for two hours at 37 °C with 8% CO_2_ to allow cell adhesion and then treated or not with 10^−8^ M BPA or BPS. The medium, supplemented or not with BPs, was changed daily for three days. On days 1 to 3, cells from three wells were collected with trypsin, centrifuged at 300 g for 10 min, resuspended in 10 mL ISOTON II diluent (Beckman Coulter, Mississauga, ON, Canada) and counted separately using a Z2 Coulter Particle Count and Size Analyzer (Beckman Coulter) [26]. A graph illustrating the numbers of cells per well as a function of time was performed to calculate the proliferation rate. Proliferation values for days 1 to 3 were established as percentages of control (i.e., untreated condition) on day 1. Therefore, the proliferation value of the control on day 1 was established as 100%. Each condition included three replicates (n = 3) for every cell line, and each experiment was repeated independently three times (N = 3).

### 2.4. Migration

UCs, RT4 and T24 cells were seeded in 12-well culture plates at 150,000 cells/well density. Cells were incubated for two hours at 37 °C with 8% CO_2_ to allow cell adhesion. Then, cells were treated or not with 10^−8^ M BPA or BPS and incubated at 37 °C with 8% CO_2_. After 72h, a scratch test was performed [27]. Briefly, a vertical scratch was performed in each well using a 200 µL pipet tip. Wells were rinsed twice to remove detached cells with 3:1 DMEM–Ham F12 medium supplemented with 0.5% fetal bovine serum clone II. Two mL of medium supplemented with 0.5% serum with or without 10^−8^ M BPA or BPS was added to each well. This low serum concentration was chosen to avoid cell proliferation and ensure the observed results are due to cell migration. Cell migration was assessed using a Zeiss Axio Imager M2 Time-Lapse microscope equipped with an AxioCam ICc1 camera (Carl Zeiss, Oberkochen, Germany). Images were processed with the AxioVision 40 V4.8.2.0 software (Carl Zeiss). Photographs were taken every hour for a total of 17 h. Analyses of closure area were measured using ImageJ software (NIH, Bethesda, MD, USA). Migration speed was calculated as the slope of the closure area (Y-axis) as a function of time (X-axis) with the formula “ax + b”, where “a” represents migration speed. The slopes of each cell line were established as percentages of control (i.e., untreated condition). Each condition included two replicates (n = 2) for every cell line, and each experiment was repeated independently three times (N = 3).

### 2.5. Flow Cytometry

RT4 and T24 cells were seeded in 12-well culture plates at 125,000 cells/well density. Cancer cells were treated with or without 10^−8^ M BPA for ten days and incubated at 37 °C with 8% CO_2_. The medium was changed three times a week. After ten days, cells were collected with trypsin, centrifuged at 300 g for 10 min and resuspended in 100 µL of PBS. Cell suspensions were individually transferred in 100 µL of 3.7% formol to fix cells, and samples were incubated at 4 °C until flow cytometry analyses. Cell samples were washed twice with PBS on the analysis day, followed by blocking using PBS-1% BSA for 45 min at room temperature. Incubation with primary antibody anti-alpha smooth muscle actin (α-SMA) coupled with FITC (1/250 dilution; Abcam) or with isotypic control FITC antibody (1/1000 dilution; Abcam) was performed at room temperature for one hour in PBS-1% BSA. Cells were washed twice with PBS-1% BSA and once with PBS only. Cells were resuspended in 100 µL PBS and samples were analyzed by flow cytometry with a FACSCalibur (Becton Dickson, San Jose, CA, USA) [28]. For each replicate, the value of the isotypic sample was subtracted from the value of the positive sample (cells incubated with antibodies). Each condition included six replicates (n = 6), and each experiment was repeated independently three times (N = 3).

### 2.6. Statistical Analysis

Graphical representation and statistical analyses were performed using Microsoft Excel (Microsoft) and GraphPad Prism Software v.9.3 (San Diego, CA, USA). The results are expressed as mean ± standard error of the mean (SEM). Statistical analyses were performed using the non-parametric tests, the Mann–Whitney test or the Kruskal–Wallis test. Normality analyses were performed for the data in Figure 1 and Figure 7. The values did not meet the required premises to assume normality, which justifies using non-parametric statistical tests. Statistical significance was established at *p* < 0.05.

## 3. Results

### 3.1. T24 Invasive Bladder Cancer Cells Exhibit an Increased Glycolytic Capacity Compared with RT4 Non-Invasive Bladder Cancer Cells

Before evaluating the impact of chronic exposure to BPs, the energy metabolism of two UC populations (UC1 and UC2) and two bladder cancer cells (RT4 non-invasive and T24 invasive) was evaluated to compare their glycolytic and mitochondrial capacities. First, UC1 and UC2 cell populations had a similar glycolytic metabolism (Figure 1A,B). However, UC2 cells exhibited higher OCR levels at basal capacity compared with UC1 cells, whereas the maximal mitochondrial capacity of UC1 and UC2 cells was not significantly different (Figure 1C,D).

Secondly, RT4 non-invasive bladder cancer cells displayed a significantly higher basal glycolysis than UC1 cells (Figure 1A). RT4 cells tended to exhibit slightly higher ECAR levels compared with UC2 cells (*p* = 0.07) (Figure 1A), whereas ECAR levels for maximal glycolytic capacity were comparable with UCs (Figure 1B). RT4 cells had higher OCR levels for basal mitochondrial respiration than both UCs (Figure 1C) but exhibited the lowest maximal mitochondrial capacity compared with the other cell lines (Figure 1D).

Thirdly, T24 invasive bladder cancer cells exhibited the highest ECAR levels for basal and maximal glycolytic capacity compared with the other cell lines (Figure 1A,B). A general representation of the energy metabolism of T24 cells shows that these cancer cells are highly glycolytic (Appendix A). T24 cells exhibit similar OCR levels for basal mitochondrial respiration compared with RT4 cells but significantly higher levels for maximal mitochondrial respiration (Figure 1C,D). As observed for the glycolytic capacity, T24 cells exhibited significantly higher levels of OCR for maximal mitochondrial respiration compared with RT4 non-invasive bladder cancer cells and UCs. Therefore, initial analyses showed that T24 cells displayed a greater glycolytic capacity than RT4 cells, whereas the two populations of UCs exhibited roughly similar energy metabolism.

### 3.2. Chronic Exposure to Physiological Concentrations of BPA or BPS Does Not Modulate the Glycolysis and Mitochondrial Respiration of Normal Urothelial Cells

Two UC populations were exposed to physiological concentrations of BPA or BPS to evaluate the impact of these endocrine disruptors on their bioenergetics. In vivo, UCs can be in contact with BPA and BPS through urine [11,29]. Chronic exposure to 10^−8^ M BPA or BPS did not seem to impact the energy metabolism of UCs. A slight but not significant decrease in ECAR levels was observed for the glycolytic capacity of both compounds (Figure 2A,B), as well as a subtly decreased OCR level in basal mitochondrial respiration when exposed to BPS (*p* = 0.12) (Figure 2C). Exposure to BPs did not affect OCR levels associated with maximal mitochondrial respiration (Figure 2D). Overall, 72 h chronic exposure to physiological concentrations of BPA or BPS did not significantly affect the bioenergetics of normal UCs (Appendix A).

### 3.3. RT4 Non-Invasive Bladder Cancer Cells Chronically Exposed to Physiological Concentrations of BPs Exhibit Increased Bioenergetics

RT4 cells were exposed to physiological concentrations of BPA or BPS to evaluate the impact of these endocrine disruptors on non-invasive bladder cancer cells. Like UCs, bladder cancer cells can be exposed to BPs through urine [11,29]. BPA exposure did not influence the ECAR levels of the basal glycolysis of RT4 cells, whereas chronic exposure of RT4 cells to 10^−8^ M BPS significantly increased the ECAR levels of basal glycolysis (Figure 3A). Chronic exposure to BPA had a non-significant effect on the ECAR levels associated with the maximal glycolytic capacity. In contrast, exposure to physiological concentrations of BPS tended to increase ECAR levels (*p* = 0.07) (Figure 3B). RT4 chronically exposed to 10^−8^ M BPA exhibited significantly increased OCR levels associated with basal and maximal mitochondrial respiration (Figure 3C,D). Although not significantly different from the control condition for mitochondrial metabolism, BPS exposure seemed to have similar biological effects to BPA, suggesting this alternative compound could have comparable effects. Therefore, chronic exposure to physiological concentrations of BPs increased the bioenergetics of RT4 non-invasive bladder cancer cells (Appendix A).

### 3.4. T24 Invasive Bladder Cancer Cells Chronically Exposed to Physiological Concentrations of BPA or BPS Exhibit an Increased Glycolytic Metabolism

T24 cells were exposed to physiological concentrations of BPA or BPS to evaluate the impact of these compounds on invasive bladder cancer cell metabolism. Chronic exposure to 10^−8^ M BPA or BPS increased the ECAR levels of basal glycolysis of invasive bladder cancer cells (Figure 4A), whereas only 10^−8^ M BPS significantly increased the ECAR levels of the maximal glycolytic capacity (Figure 4B). Although not significant, BPA seemed to slightly increase the ECAR levels of the maximal glycolysis of T24 cells (*p* = 0.096) (Figure 4B). Chronic exposure to BPA and BPS did not affect the OCR levels associated with the basal and maximal mitochondrial respiration of T24 cells (Figure 4C,D). As previously observed with RT4 cells, BPA and BPS seemed to have similar effects on the energy metabolism of invasive bladder cancer cells (Appendix A). Overall, T24 invasive bladder cancer cells exhibited an increased glycolytic metabolism when chronically exposed to physiological concentrations of BPs.

### 3.5. Chronic Exposure to Physiological Concentrations of BPA or BPS Increases the Proliferation Rate of RT4 Non-Invasive Bladder Cancer Cells and Induces an Initial Boost of Proliferation for UCs and T24 Cells

The proliferation rate of urothelial, RT4 and T24 cells was evaluated for three days under chronic exposure to 10^−8^ M BPA or BPS. First, UCs exposed to BPA or BPS exhibited a significantly higher cell number on days 1 to 3 compared with the control (Figure 5A). However, it is possible to observe that, following this initial increase, the proliferation rate seemed to stabilize, suggesting that the proliferation rate could be enhanced in the first 24 h of BPA or BPS exposure, resulting in an initial boost. Secondly, RT4 non-invasive bladder cancer cells chronically exposed to 10^−8^ M BPA or BPS exhibited a significantly higher proliferation rate on day 3 than the control (Figure 5B). However, exposure to BPs did not impact RT4 cell proliferation on days 1 and 2. Thirdly, chronic exposure of T24 cells to BPA or BPS did not affect the proliferation rate on days 2 and 3. However, exposure to BPA significantly increased the proliferation rate of T24 cells on day 1 (Figure 5C), but its clinical impact is probably not significant. This observation could be associated with an initial increase in proliferation due to BPA exposure since the initial effect observed following BPA or BPS exposure was not maintained on days 2 and 3. Therefore, chronic exposure to physiological concentrations of BPA or BPS increased the proliferation rate of RT4 cells and induced an initial boost of proliferation in the first 24 h for normal UCs and T24 cells.

### 3.6. Chronic Exposure to Physiological Concentrations of BPA or BPS Decreases the Migration of Normal Urothelial Cells While Increasing the Migration Speed of Bladder Cancer Cells

The migration of urothelial, RT4 and T24 cells was evaluated under chronic exposure to 10^−8^ M BPA or BPS. On the one hand, chronic exposure to physiological concentrations of BPA significantly decreased the migration speed of UCs. In contrast, BPS exposure only resulted in a slight reduction in migration (*p* = 0.056) (Figure 6A). On the other hand, 10^−8^ M BPA tended to increase the migration speed of RT4 non-invasive bladder cancer cells (*p* = 0.06), but BPS did not affect the migration of RT4 cells (Figure 6B). BPA and BPS significantly increased the migration speed of T24 invasive bladder cancer cells (Figure 6C). Therefore, chronic exposure to physiological concentrations of BPA or BPS decreased normal UCs while increasing the migration of T24 invasive bladder cancer cells.

### 3.7. RT4 Non-Invasive Bladder Cancer Cells Chronically Exposed to Physiological Concentrations of BPA Exhibit an Increased Expression of α-SMA Expression

RT4 and T24 cancer cells were chronically exposed to 10^−8^ M BPA and analyzed by flow cytometry to evaluate the impact of this endocrine disruptor on the expression of alpha-smooth muscle actin (α-SMA), which can be used as a cell marker for the invasive potential of cancer cells [30]. Compared with T24 invasive bladder cancer cells, RT4 non-invasive cancer cells expressed significantly lower levels of α-SMA (Figure **7**). However, when chronically exposed to 10^−8^ M BPA, RT4 cells exhibited an increased expression of α-SMA. Furthermore, α-SMA levels of RT4 cells exposed to BPA were not significantly different from the α-SMA levels of T24 cells. The chronic exposure of T24 cells to 10^−8^ M BPA did not substantially affect the α-SMA expression levels. See Appendix A for an example of gating analysis. Overall, chronic exposure of RT4 non-invasive bladder cancer cells to physiological concentrations of BPA increased the expression of α-SMA.

## 4. Discussion

Since the effects of BPs on bladder cancer have not yet been established, the impact of chronic exposure to physiological concentrations of BPA or BPS on normal urothelial cells and non-invasive and invasive bladder cancer cells was studied. Therefore, the effects of BPs on energy metabolism, proliferation, migration and α-SMA expression were examined.

Before evaluating the impact of chronic exposure to physiological concentrations of BPs, the energy metabolism of UCs and RT4 non-invasive and T24 invasive bladder cancer cells was evaluated to compare their glycolytic and mitochondrial capacities. The two different populations of normal UCs (UC1 and UC2) showed similar levels of energy metabolism, and RT4 non-invasive bladder cancer cells displayed higher basal glycolysis and mitochondrial respiration than UCs. In contrast, T24 invasive bladder cancer cells exhibited a greater glycolytic capacity than normal UCs and RT4 non-invasive bladder cancer cells.

Cancer cells typically exhibit an increased metabolic rate due to enhanced physiological activity, characterized by an uncontrolled proliferation rate [31]. Cancer cells are characterized by increased glycolytic metabolism, at the expense of mitochondrial respiration, even in the presence of oxygen and functioning mitochondria [31]. This metabolic switch is called the Warburg effect, a phenomenon that allows cancer cells to have an adapted energy metabolism to support cell growth and proliferation and promote cell invasion [31]. The increased intake of glucose results in a high synthesis of pyruvate. Since the glycolytic rate is superior to the mitochondrial capacity, the excess pyruvate is converted to lactate through the enzyme lactate dehydrogenase [32].. The increased use of the glycolytic pathway results in enhanced lactate production, acidifying the extracellular microenvironment. Lactate can inhibit immune cells, such as monocytes, that are responsible for eliminating unhealthy cells, including cancer [33]. The accumulation of lactate also reorganizes the extracellular matrix, which could promote tumour invasion [22]. However, this enhanced glycolysis was only observed in the RT4 cells for basal glycolysis, not maximal glycolysis. RT4 cells had a significantly higher basal glycolytic metabolism than UC1, but not UC2, and had a similar maximal glycolytic metabolism compared with both UCs. The basal mitochondrial capacity of RT4 cells was increased compared with UCs. The absence of an enhanced maximal glycolytic capacity when compared with T24 cells could be because RT4 cells are non-invasive, therefore, are less prone to produce an excess of lactate to invade the extracellular matrix.

Furthermore, RT4 cells had a slower doubling rate than T24 cells. The comparison of the proliferation rate of untreated RT4 and T24 cells demonstrates an important difference between both cell lines. RT4 cells have a doubling time of 37 to 66 h [34,35], whereas T24 cells are reported to have a doubling time of 19 to 24 h [34,35,36]. Therefore, RT4 non-invasive bladder cancer cells have a lower level of physiological activity, which could explain the absence of metabolic switch observed. Finally, UCs exhibited a similar energy metabolism even though UC2 cells had higher levels of basal mitochondrial respiration. This difference could be due to individual variations associated with genetics or environmental factors, such as drugs, chemical exposure and health [37].

In vivo, the basal layer’s UCs are protected from exposure to urine and its contaminants, such as BPs. However, cancer cell growth can alter the impermeability of the urothelium by disrupting cell–cell adhesion [38], thus resulting in the exposure of UCs surrounding the tumour and underlying UCs to urine and potentially BPs. UCs can also be exposed to BPs through the bladder vascular system perfusing the bladder stroma [39]. BPA and BPS had a weak effect on the bioenergetics of UCs. Although not significant, this tendency could be associated with inhibiting certain enzymes related to cell metabolism by BPs. BPA has been shown to inhibit metabolizing enzymes, such as cytochromes P450, glucose transporters and enzymes from the electron transport chain, that can impact some main metabolic pathways, for example, mitochondrial respiration [40,41]. It is, therefore, essential to confirm the metabolic analyses with physiological parameters to evaluate the functional impact of BPA or BPS on UCs.

The alteration in the energy metabolism of RT4 cells after chronic exposure to BPs could be associated with the multiple binding capacities of BPA and BPS. BPA was shown to bind to ERs with a median inhibitory concentration (IC_50_) value of 3.3–73 nM [42], which corresponds to the range of concentrations of BPs found in urine (1–100 nM) [11]. Therefore, the increased energy metabolism observed could be related to the enhanced physiological activity of RT4 cells following exposure to BPs. Although the impact of BPS on mitochondrial capacity was not significantly different from the control condition, chronic exposure to this compound seems to induce similar effects to BPA. These results need to be correlated through the impact of BPA or BPS on the physiological activity of RT4 cells. The relevance of these results depends on the associated biological consequences, for example, proliferation and migration.

Thirdly, T24 invasive bladder cancer cells chronically exposed to physiological concentrations of BPA or BPS did not seem to impact the mitochondrial capacities but exhibited an increased glycolytic metabolism. The increased glycolytic capacity observed in T24 cells following chronic exposure to BPA or BPS could lead to higher lactate production and enhance the acidification of the tumour microenvironment. Although T24 cells are invasive cancer cells, chronic exposure to BPA or BPS could accentuate the consequences of matrix acidification [31], further inhibiting the local immune cells [33], thus facilitating invasion and metastases formation through a major matrix remodelling.

The increased proliferation rate of RT4 non-invasive cancer cells chronically exposed to physiological concentrations of BPA or BPS could suggest a more aggressive phenotype. Indeed, studies have shown that BPA could promote cell proliferation by binding to the GPER in breast cancer cells, which activates the phosphoinositide 3-kinase (PI3K)/protein kinase B (Akt)/mammalian target of rapamycin (mTOR) signalling pathway [43]. Furthermore, BPA can bind to the ERRγ of endometrial cancer cells, which can activate the epidermal growth factor receptor (EGFR)/extracellular signal-regulated kinase (ERK) pathway associated with proliferation [44]. On the other hand, BPS has also been shown to promote cell proliferation by altering the PI3K/Akt signalling pathway in breast cancer cells [45]. BPA or BPS exposure to T24 invasive bladder cancer cells did not impact the proliferation rate in the long term. Still, BPA induced an increased proliferation in the first 24 h. BPA can alter multiple signalling pathways associated with proliferation, which could explain the increased proliferation rate. However, this difference was not observed on days 2 and 3, which could be explained by the already high proliferation rate of T24 cells, therefore obscuring the impact of BPs in the long term.

Next, the impact of chronic exposure to BPA or BPS on cell migration was evaluated. UCs chronically exposed to physiological concentrations of BPA or BPS exhibited decreased migration. This observation could impact the urothelium tissue repair in the case of disease or bladder injury. Cell migration is essential in wound healing to allow epithelialization and wound closure [46]. The decreased migration capacity of UCs could result in a slower wound closure of the bladder wall, increasing the exposure of underlying tissues to the toxic substances found in urine, such as urea and carcinogens [47]. Chronic exposure to BPA or BPS did not impact RT4 cell migration. RT4 cells have a low migration capacity as these cells form and stay in clusters, resulting in minimal migration. Therefore, the impact of BPs could potentially be obscured by the fact that RT4 cells simply do not tend to migrate. However, chronic exposure to BPA or BPS increased the migration of T24 invasive bladder cancer cells. On the one hand, BPA has been shown to increase the migration of lung cancer cells through the GPER/EGFR/ERK1/2 signalling pathway [48] and increase the migration of triple-negative breast cancer cells through ERRγ [7]. Studies by Derouiche et al. demonstrated that exposure to BPA can promote the cell migration of prostate cancer cells through the modulation of ion channel protein expression associated with calcium entry [8]. On the other hand, exposure to BPS has been shown to promote the migration of human non-small cell lung cancer cells through ERK1/2, mediated by the transforming growth factor β (TGF-β)/Smad-2/3 signalling pathway [11]. Deng et al. have reported that BPS can also promote the migration of triple-negative breast cancer cells in vitro through the GPER/Hippo signalling pathway, resulting in the activation and nuclear accumulation of yes-associated protein (YAP), thus upregulating downstream genes [49]. Therefore, these results concur with the literature and the metabolic analyses, demonstrating an enhanced energy metabolism in T24 cells when exposed to BPA or BPS.

In brief, chronic exposure to physiological concentrations of BPs tended to decrease the energy metabolism of UCs, increase their proliferation in the first 24 h and decrease their migration. In addition, exposure to BPs increased the energy metabolism and the proliferation of non-invasive RT4 bladder cancer cells but had little to no effect on their migration. Finally, exposure to BPs increased the glycolytic capacity and the migration of invasive T24 bladder cancer cells but did not impact their mitochondrial respiration or proliferation (Table 1). These results suggest that chronic exposure to BPA and BPS alters the energy metabolism and behaviours of UCs and non-invasive and invasive bladder cancer cells, which could potentially promote bladder cancer progression.

The impact of chronic exposure to BPA on the α-SMA expression of RT4 and T24 cells was established. α-SMA can be used as an epithelial–mesenchymal transition (EMT) marker, during which α-SMA expression is increased [30]. When cancer cells transit from a non-invasive to an invasive and metastatic phenotype, they undergo various steps, including EMT [50]. Increased α-SMA expression is associated with an invasive phenotype and a greater capacity to produce metastases [30], therefore resulting in a poorer prognosis in multiple cancers such as lung [51] and breast cancer [52]. Consequently, invasive cancer cells tend to express higher levels of α-SMA than non-invasive cancer cells [30], allowing α-SMA expression to be used as a cell marker for cancer cell aggressiveness. The results show that RT4 non-invasive bladder cancer cells expressed significantly lower levels of α-SMA when compared with T24 invasive bladder cancer cells. Chronic exposure of RT4 cells to a physiological concentration of BPA increased the expression of α-SMA. Chronic exposure to BPA could, therefore, enhance the aggressiveness of non-invasive cancer cells, thus promoting the transition from a non-invasive to an invasive phenotype. Clinically, these results suggest that chronic exposure to BPs could promote bladder cancer recurrence and progression.

This study is not without limitations. First, the inability to ensure a bisphenol-free control represents a limitation [53]. Since bisphenol-based plastics are abundantly used in laboratory equipment and scientific instruments, it is challenging to avoid external bisphenol contamination. However, there were still significant differences between controls and the BPA or BPS conditions. Second, using an in vitro 2D cell culture is a notable limitation. Using a 3D bladder cancer model [54] would better represent the physiological impact of chronic exposure to BPA or BPS on UCs and bladder cancer tumours. RT4 and T24 cells were selected in this study for their ability to form spheroids, as we are planning to further investigate the effects of BPs using our 3D bladder cancer model [54].

Measuring the presence of BPs in different stages of bladder cancer samples would be an interesting perspective for this study to determine if advanced bladder cancers exhibit higher levels of BPs than low-grade bladder cancer patients. Furthermore, evaluating the impact of BPs’ metabolites on bladder cancer progression would be valuable. A significant proportion of BPs is metabolized and excreted through urine [55]. However, studies have shown that these metabolites can remain active in the body [56], suggesting they could also potentially affect cancer development. Unfortunately, the impact of BPs’ metabolites has never been investigated in bladder cancer.

## 5. Conclusions

Few studies have evaluated the potential impact of chronic exposure to BPs on bladder cancer, despite the presence of these endocrine disruptors in human urine and the accumulating literature demonstrating their pro-tumorigenic capacities. This study has brought valuable insights into the effects of chronic exposure to a physiological concentration of BPA or BPS on bladder cancer progression. The impact of these BPs on the energy metabolism, proliferation, migration and α-SMA expression of normal UCs and RT4 non-invasive and T24 invasive bladder cancer cells was established. The results show that chronic exposure to BPA or BPS increases the proliferation rate of UCs while decreasing their migration, which could result in hyperplasia and altered wound healing capacities. Exposure to BPA or BPS also increases cancer cells’ energy metabolism and physiological activity, particularly the metabolism, proliferation and α-SMA expression of RT4 non-invasive bladder cancer cells, which could promote bladder cancer progression from a non-invasive to an invasive phenotype. The ubiquitous and continuous exposure to these endocrine disruptors, through food and the environment, could have a meaningful clinical impact on bladder cancer recurrence, progression and patient prognosis.

## Figures and Tables

**Figure 1 cancers-14-04011-f001:**
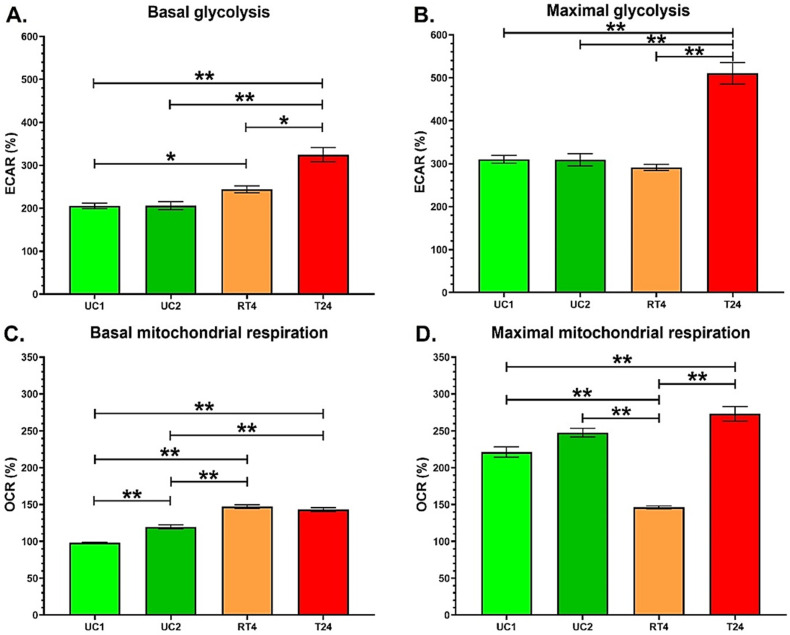
T24 invasive bladder cancer cells exhibit an increased glycolytic capacity compared with RT4 non-invasive bladder cancer cells. (**A**,**B**) ECAR and (**C**,**D**) OCR were determined using the XFe96 Extracellular Flux Analyzer in two populations of normal urothelial cells (UC1 and UC2) and non-invasive (RT4) and invasive (T24) bladder cancer cells to establish (**A**) basal glycolysis, (**B**) maximal glycolytic capacity, (**C**) basal mitochondrial respiration and (**D**) maximal mitochondrial capacity. Data are presented as the mean ± SEM and displayed as percentages of UC1 acting as control (n = 10, N = 4). A baseline (100%) was established before injections (see Appendix A). * *p* < 0.05, ** *p* < 0.01 by Kruskal–Wallis test.

**Figure 2 cancers-14-04011-f002:**
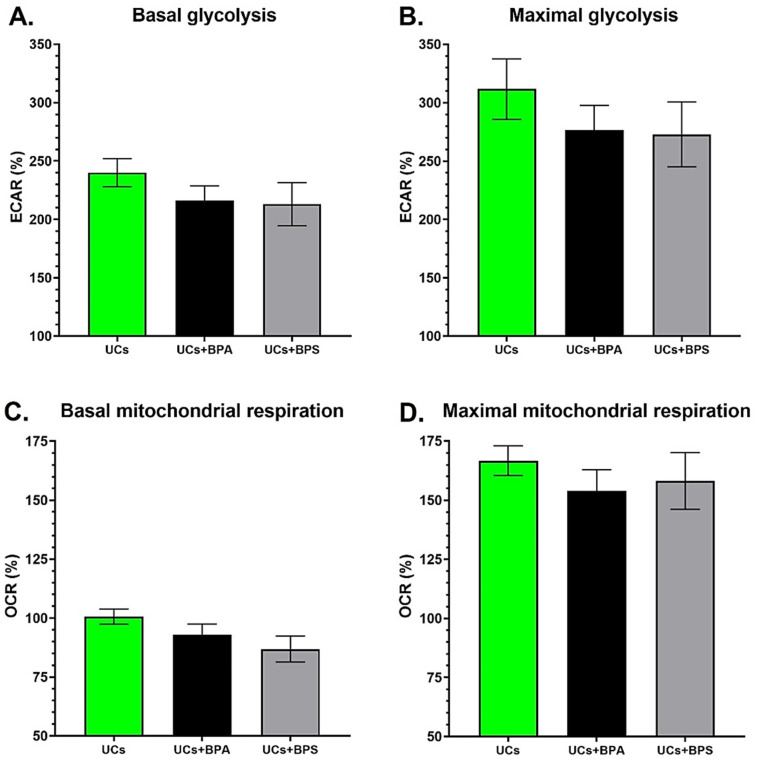
Chronic exposure to physiological concentrations of BPA or BPS has no significant effect on the glycolysis and mitochondrial respiration of normal urothelial cells. (**A**,**B**) ECAR and (**C**,**D**) OCR were determined using the XFe96 Extracellular Flux Analyzer for normal urothelial cells (UCs) with or without chronic exposure to physiological concentrations of BPA or BPS to establish (**A**) basal glycolysis, (**B**) maximal glycolytic capacity, (**C**) basal mitochondrial respiration and (**D**) maximal mitochondrial capacity. Analyses represent the results for two populations of normal urothelial cells (UC1 and UC2). Data are presented as the mean ± SEM and displayed as percentages of controls (i.e., untreated condition) (n ≥ 3, N = 4). A baseline (100%) was established before injections (see Appendix A). *p* < 0.05 by Mann–Whitney test.

**Figure 3 cancers-14-04011-f003:**
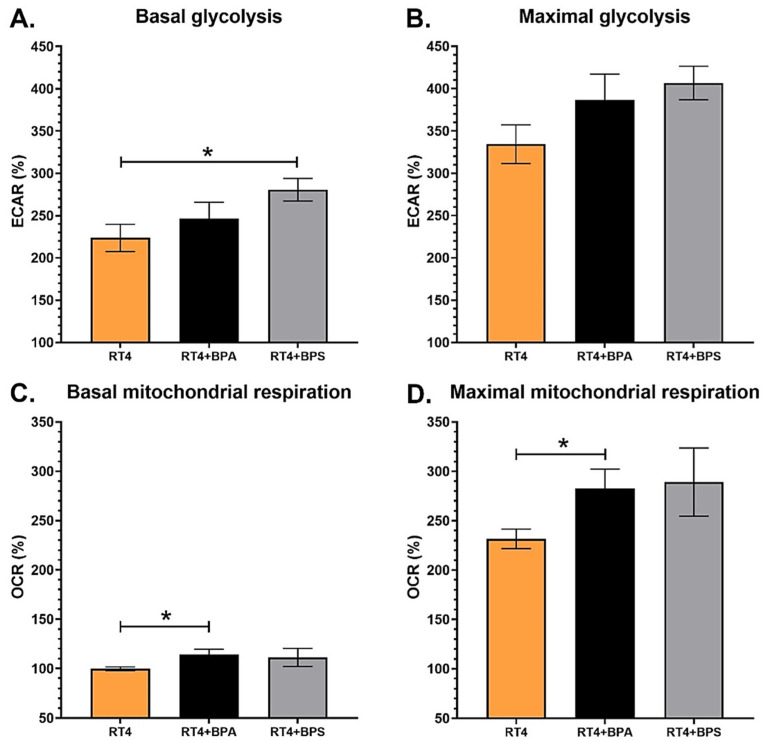
RT4 non-invasive bladder cancer cells chronically exposed to physiological concentrations of BPs exhibit increased bioenergetics. (**A**,**B**) ECAR and (**C**,**D**) OCR were determined using the XFe96 Extracellular Flux Analyzer for RT4 non-invasive bladder cancer cells with or without chronic exposure to physiological concentrations of BPA or BPS to establish (**A**) basal glycolysis, (**B**) maximal glycolytic capacity, (**C**) basal mitochondrial respiration and (**D**) maximal mitochondrial capacity. Data are presented as the mean ± SEM and displayed as percentages of controls (i.e., untreated condition) (n ≥ 3, N = 3). A baseline (100%) was established before injections (see Appendix A). * *p* < 0.05 by Mann–Whitney test.

**Figure 4 cancers-14-04011-f004:**
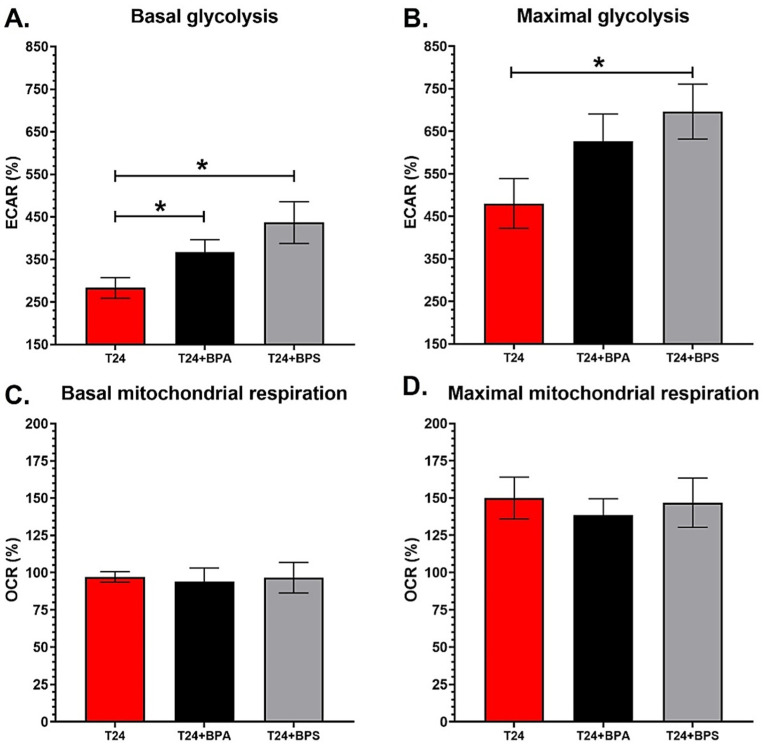
T24 invasive bladder cancer cells chronically exposed to BPA or BPS physiological concentrations exhibit an increased glycolytic metabolism. (**A**,**B**) ECAR and (**C**,**D**) OCR were determined using the XFe96 Extracellular Flux Analyzer for T24 invasive bladder cancer cells with or without chronic exposure to physiological concentrations of BPA or BPS to establish (**A**) basal glycolysis, (**B**) maximal glycolytic capacity, (**C**) basal mitochondrial respiration and (**D**) maximal mitochondrial capacity. Data are presented as the mean ± SEM and displayed as percentages of controls (i.e., untreated condition) (n ≥ 3, N = 3). A baseline (100%) was established before injections (see Appendix A). * *p* < 0.05 by Mann–Whitney test.

**Figure 5 cancers-14-04011-f005:**
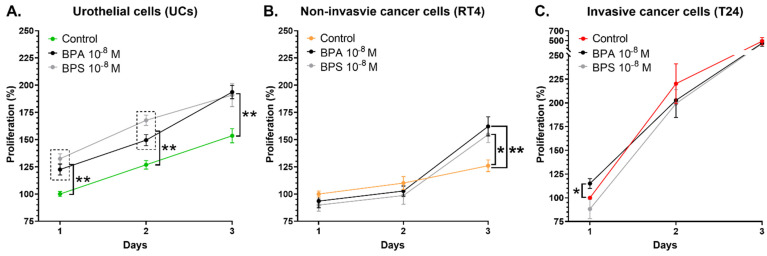
Chronic exposure to physiological concentrations of BPA or BPS increases the proliferation rate of RT4 non-invasive bladder cancer cells and induces an initial boost of proliferation for UCs and T24 cells. The proliferation rate of (**A**) normal urothelial cells (UCs), (**B**) RT4 non-invasive bladder cancer cells and (**C**) T24 invasive bladder cancer cells was established over three days with or without chronic exposure to physiological concentrations of BPA or BPS. The proliferation rate is illustrated by the number of cells as a function of time. Data are presented as the mean ± SEM and displayed as percentages of controls at day 1 (i.e., untreated condition) (n = 3, N = 3). * *p* < 0.05, ** *p* < 0.01 by Mann–Whitney test.

**Figure 6 cancers-14-04011-f006:**
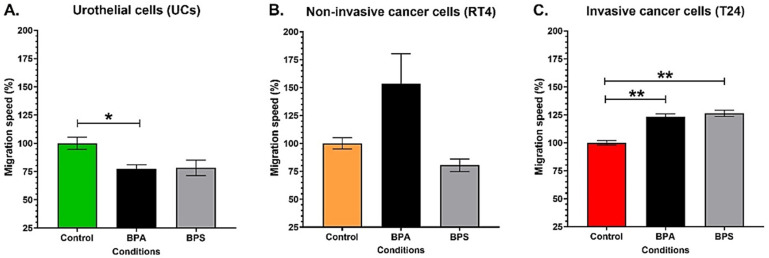
Chronic exposure to physiological concentrations of BPA or BPS decreases the migration speed of normal urothelial cells while increasing the migration speed of bladder cancer cells. The migration speed of (**A**) normal urothelial cells (UCs), (**B**) RT4 non-invasive bladder cancer cells and (**C**) T24 invasive bladder cancer cells was evaluated by time-lapse microscopy with or without chronic exposure to physiological concentrations of BPA or BPS. Data are presented as the mean ± SEM and displayed as percentages of controls (i.e., untreated condition) (n = 2, N = 3). The 100% migration value of UCs represents a mean of 12,06 cm^2^/h, for RT4 1,68 cm^2^/h and for T24 4,77 cm^2^/h. * *p* < 0.05, ** *p* < 0.01 by Mann–Whitney test.

**Figure 7 cancers-14-04011-f007:**
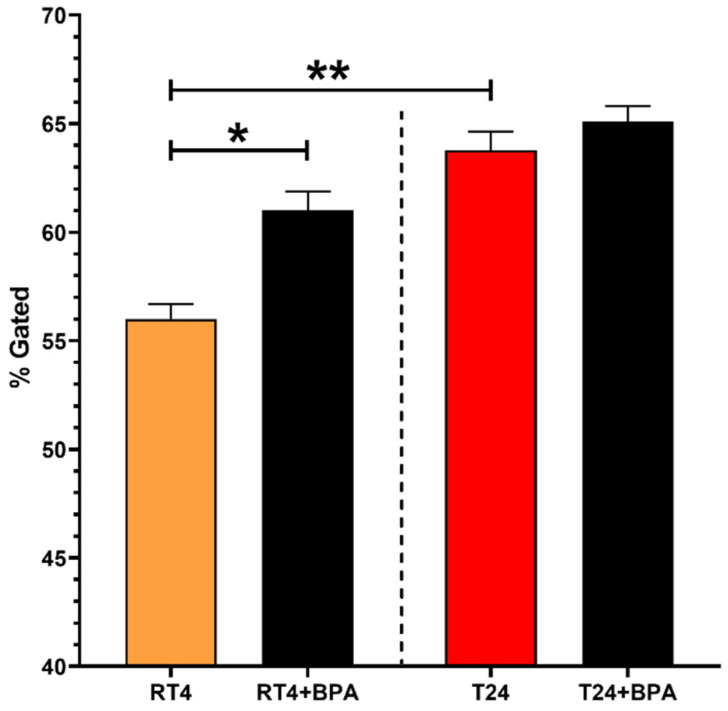
RT4 non-invasive bladder cancer cells chronically exposed to BPA’s physiological concentrations exhibit an increased α-SMA expression. The expression of α-smooth muscle actin (α-SMA) by RT4 non-invasive and T24 invasive bladder cancer cells was measured by flow cytometry with or without chronic exposure to physiological concentrations of BPA. Data are presented as the mean ± SEM (n = 6, N = 3). * *p* < 0.05, ** *p* < 0.001 by Kruskal–Wallis test.

**Table 1 cancers-14-04011-t001:** Impact of chronic exposure to physiological concentrations of BPA and BPS on the energy metabolism, proliferation and migration of UCs and RT4 non-invasive and T24 invasive bladder cancer cells. UCs are generally negatively affected by exposure to BPs, illustrated by the decreased bioenergetics and migration. Conversely, exposure to BPs generally enhances the bioenergetics, proliferation and migration of cancer cells.

Cell Types	Parameters		BPA	BPS
UCs	Glycolysis	Basal	↓	↓
		Maximal	↓	↓
	Mitochondrial respiration	Basal	Ø	↓
		Maximal	Ø	Ø
	Proliferation		↑	↑
	Migration		↓↓	↓
RT4 cells	Glycolysis	Basal	Ø	↑↑
		Maximal	Ø	↑
	Mitochondrial respiration	Basal	↑↑	↑
		Maximal	↑↑	↑
	Proliferation		↑↑	↑↑
	Migration		↑	Ø
T24 cells	Glycolysis	Basal	↑↑	↑↑
		Maximal	↑	↑↑
	Mitochondrial respiration	Basal	Ø	Ø
		Maximal	Ø	Ø
	Proliferation		Ø	Ø
	Migration		↑↑	↑↑

Legend: Ø = No impact; ↑/↓ = Tendency; ↑↑/↓↓ = Significant.

## Data Availability

The data presented in this study are available on request from the corresponding author.

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
