# Peer review of "Bisphenols A and S Alter the Bioenergetics and Behaviours of Normal Urothelial and Bladder Cancer Cells"

_cancers, 2022, doi:10.3390/cancers14164011_

Round 1

Reviewer 1 Report

Dear authors congratulation for your hard work. It is common knowledge that even though the last decades science had tremendous breakthroughs, still we miss numerous data from understanding, predicting and treating cancer, in every form. Once in a while, research such yours add valuable data in scientists effort in making this happen.

If I may I want to share some thoughts and remarks uppon your study.

1. Your control sample has been obtained by 2 healthy volunteers. It would be better not to reffer to them as patients and to prove the abscence of cancer by including a pathology report.

2. The control group in order to be healthy should meet some kind of criteria, for instance non smokers, no occupational hazards etc. Please clarify.

3. Why did you use RT4 non-invasive bladder cancer cells (ATCC HTB-2) and T24 invasive bladder 109 cancer cells (ATCC HTB-4)? It would be best if you clarify the reasons you have chosen them.

4. As you have mentioned there are certain limitations in your study. Obviously it is very difficult to have a BPs free control sample.How about measuring the extent of BPs exposure in non invasive and invasive bladder cancer samples?

Again congratulations for your hard work. I am looking forward to your new step.

Reviewer 2 Report

Ève Pellerin et al. investigated very interesting research titled “Bisphenols A and S alter the bioenergetics and behaviours of normal urothelial and bladder cancer cells”. The manuscript is well written overall. Before it may be considered for publication, the following revision needs to be made. 

1.      The methodology section of the abstract lacked a lot of detail. I read more about the background of the study and its findings. Even the conclusion needs more information regarding the research's future directions.

2.      Make the introduction concise, particularly paragraphs 3. It's too long, and I've given up paying attention to it because of that.

3.      The materials and methods have very few references. Only two references were included in the methods section. More references are required to back up the methods used in this study.

4.      Is the 5 g/ml insulin is accurate?

5.      The results section is presented well. Yet I believe the discussion portion is rather lengthy (almost 3 pages). This will turn off readers' interest in it. Reduce it. The author should offer a critical justification of the findings rather than more literatures.

6.      All of the figures are quite legible and informative. However, Figure 7's bars have the grid lines, which is to be removed, making them empty. The figure 7's colours also contrast with one another.

7.      Delete I, we, our throughout the manuscript.

Round 2

Reviewer 2 Report

According to the suggestions, the authors substantially revised the manuscript. As a result, I recommend that this be taken into consideration for publication in Cancers Journal in its current form.